# A Power Demand Analytical Model of Self-Propelled Vessels

**Javier Zamora**

Navalytica, 8 The Green, Dover, DE 12872, USA; javier@navalytica.com

**Abstract:** The article herein presents a closed-form mathematical equation by which it is possible to estimate the propulsion power demand of ships as a function of the propeller parameters and total Resistance. The validation of the derived model is conducted by use of the Series 60 Model data and of the Korea Research Institute of Ships and Ocean Engineering (KRISO) Very Large Crude-oil Carrier 2 (KVLCC2) data. In all the cases tested, the derived model explained more than 99.9% of the data variability. Furthermore, the paper describes a practical method for quantifying changes in hull and propeller performance and provides an application example.

**Keywords:** vessel performance prediction; vessel performance evaluation; vessel power prediction; biofouling; ship performance

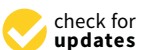



## 1. Introduction

In January 2014, the International Maritime Organization (IMO) introduced amendments to MARPOL Annex VI "Regulations for the prevention of air pollution from ships" [1] to quantify the ratio of the environmental costs to the transport capacity-mile achieved by ships through the mandatory Energy Efficiency Design Index (EEDI) for new ships, and the Ship Energy Efficiency Management Plan (SEEMP) for all ships.

More recently, the IMO adopted the technical measure Energy Efficiency Existing Ship Index (EEXI) [2], mandatory for all ships after January 2023, as well as provided guidelines on survey and certification of the EEXI [3]. In addition, guidelines on the shaft/engine power limitation system to comply with the EEXI requirements and use of a power reserve were adopted as a resolution [4], providing shipowners the option to improve the EEXI without compromising safety.

Beyond the regulatory framework, private initiatives, such as the Poseidon Principles, appeared to promote international shipping decarbonization. The Poseidon Principles were adopted by major banks in the shipping industry to integrate the ship operational performance [5] into the lending decision process.

In parallel, the shipping industry currently operates in an economic sphere in which the markets of the goods transported, as well as the particularities of the shipping markets, determine operating profiles, costs, and prices [6]. In addition, strategic investments oriented to increase fuel efficiency face the intricacies of the interactions between shipowner, charterers, and ship managers [7].

The increase of voyage costs as a percentage of revenue, either due to the rise of fuel costs or the reduction of freight rates due to the overcapacity of ships, makes fuel efficiency a key element in the ability of a shipowner to remain competitive. Thus, operational decisions by ship owners and managers tend to be considered in terms of fuel reduction.

To realize savings, assess investment risks, and remain competitive in tough financial and regulatory times, changes in performance must be quantified for their conversion to a monetary impact. As described by Armstrong [8], quantification is a significant aspect of the development of optimization initiatives.

The arrival of data acquisition systems and improvement of sensor accuracy made available large amounts of operational vessel sailing data to stakeholders already incen-

tivized due to fuel costs and international regulation to find ways to reduce operational costs by increasing vessel efficiency.

This document introduces a mathematical model with useful applicability with vessel operational data. The derivation of the model will be conducted in two steps. First, a set of equations will be derived that accurately characterize the open-water characteristics of a propeller. Then, these equations will be extended to cover full-scale vessels. The validation of the model will be performed by correctly predicting the shaft power demand of several Series 60 Models, as well as the KRISO Very Large Crude-oil Carrier 2 (KVLCC2).

## 2. State of the Art of Vessel Performance Modeling

According to ISO 19030 [9], vessel performance refers to the relationship between the condition of hull and propeller and the power required to move the ship at a given speed. Current approaches to ship performance modeling can be broadly categorized into theoretical, statistical, and hybrid methods:

Theoretical models are based on model tests that determine calm water resistance on top of which is considered the added resistance due to wind, waves, current, and fouling. The calm water resistance is the sum of frictional, residual, and air resistance. A standard model–ship correlation line (1957 ITTC) accounts for scale effects. The exact total resistance calculation method is outlined in the 1978 ITTC Performance Prediction Method [10].

Hansen [11] includes theoretical models for added resistance in wind, waves, steering, and shallow water. Eljart [12] includes the effect of sea state, wind, course-keeping, and shallow water. Hansen [11] corrects for wind/weather to calculate the power demand at a reference speed and draft to quantify the fouling effect. In addition, there are some semi-empirical models, acceptable from an initial design perspective, such as those by Holtrop and Mennen [13], Guldhammer and Harvald [14,15], Hollenbach [16], and Taylor and Gertler [17], Harvald and Hee [18].

However, the underlying formulae in all theoretical models have assumptions and associated uncertainties. Logan [19] indicates that many of the theoretical models that measure the ship's resistance remain un-validated in the scenario in which they are applied. In addition, the hull and propeller fouling creates difficulties for validating models as each added resistance cannot be attributed to its source. The weather conditions limit opportunities for sea trials validation since calm conditions are needed [20].

Further, full validation requires a large dataset that represents a wide range of ship operating conditions which may take many years to accumulate. There are also inconsistencies surrounding which specific added resistance factors should be included. In addition, no described method accounts for interaction effects between each component of added resistance.

Statistical and Machine Learning Models. Bocchetti et al. [21] proposed a statistical approach founded in multiple linear regression that allows both pointwise and interval predictions of fuel consumption.

Brandsaeter and Vanem [22] applied regression models to predict a ship's speed using a set of 18 vessel parameters collected from high-frequency sensors over 3 months. The goal of outperforming the Admiralty coefficient formula ($C_{ADM} = \Delta^{2/3}V^3/P_S$) was not achieved for the complete range of operational speeds. Perera and Mo [23,24] proposed a three-step procedure for operational data processing: sensor faults detection, data classification, and data compression using Principal Components Analysis and Gaussian Mixture models, but no quantitative metrics were published.

Bui and Perera [25] proposed a data analytics framework for ship performance monitoring under localized operational conditions with data anomaly detection based on Singular Value Decomposition (SVD) and clustering of the operational conditions based on Gaussian Mixture Models (GMM).

Ahlgren and Thern [26] relied on an unsupervised machine learning algorithm to predict ship fuel consumption. Their best-performing model achieved accuracy similar to previous researchers but with a lower number of used features. Soner et al. [27] developed ship propul-

sion models based on shrinkage models such Ridge and Lasso over high-frequency data. They utilized the same dataset as Petersen et al. [28] and reported similar accuracy.

Wang et al. [29] also used a Lasso regression to model the fuel consumption of containerships from a dataset that included 97 vessels, significantly improving the accuracy of predictions. Gkerekos et al. [30] developed a three-step process: data pre-processing, the training of a family of regression models, and selection of the best performing over the test set.

Farag and Olçer [31] combined high-frequency data with weather data with an ANN and a multi-regression model to predict a Very Large Crude Carrier (VLCC) tanker's brake power and specific fuel oil consumption, achieving high accuracy (99.6%) predicting the same dataset used to train the model. Gkerekos and Lazakis [32] combined a deep-neural network prediction model with a weather routing algorithm. Anomalies in the ship dataset were filtered by applying a $\pm 3\sigma$ cut-off value in each parameter.

Coraddu et al. [33] used random forests as a feature selection strategy. Coraddu et al. [34] blended auto-logged and Automated Identification System (AIS) data from a research vessel to train Support Vector Machines and k-nearest neighbors to classify the vessel's hull and propeller condition as "clean" or "fouled". Coraddu et al. [35] used a large dataset obtained from onboard sensors of two Handymax chemical/product tankers to develop the ships' digital twin with Neural Networks to estimate the speed loss due to marine fouling, outperforming the ISO 19030 standard approach.

Hu et al. [36] built an ensemble of different decision trees ensemble methods combined with a linear regression model to predict the daily fuel consumption of a container ship.

Kim et al. [37] developed a prediction model based on an Artificial Neural Network (ANN) to predict the fuel consumption of a 13,000 TEU container ship.

Aldous [38] and Themelis et al. [39] compared data from noon reports (NR) to continuous monitoring (CM) data, concluding that there is a significant reduction of uncertainty by using CM.

Zhu et al. [40] addressed the issue of heterogeneous data input from noon reports and continuous monitoring employing a series of moving overlapped frames to merge different frequency data and then built predictive models based on linear regression, support vector regression, and artificial neural network.

Statistics and machine learning models make it difficult to detect the significance of input variables and to understand the inner consistency between parameters. In addition, these approaches require the dataset to be an unbiased sample, and this seldom happens because operational constraints produce preferred speeds, drafts, and trims in vessel operational sailing datasets.

Hybrid Models. Telfer [41] assumed a linear relationship between the torque coefficient and the slip and proposed the Generalized Power Diagram (GPD), which relates power, ship speed, propeller revolutions, and slip for a particular wake fraction in one diagram. The generalized power diagram can be derived either from speed trials [41,42] or propeller open-water characteristics from model tests [41,43]. Bonebakker [44] applied Telfer approach over operational sailing data to analyze the performance of a tanker, and extends [45] the model to cover draft changes. Silovic and Fancev [46] derived Telfer's model from the open-water characteristics of the scaled model propeller.

Journée, Rijke et al. [47] developed a hybrid model of a ship's fuel consumption. Measured signals were used to adjust the coefficients of the hydrodynamic model over various draft, trim and speed combinations in a calm sea, to predict vessel speed, power, and fuel consumption. Predictions were found to be poor in bad weather conditions assumed due to inaccurate weather measurements.

Munk [48] described a commercial model that predicts hull fouling using weekly recordings of performance data taken with constant navigation, calm weather, and controlled draft. The added resistance due to fouling was obtained by comparing between observed values and the model output. The model is based on first principles and approximation formulae with empirical constants, although the accuracy of results was not disclosed.

Leifsson, Sævarsdóttir et al. [49] developed a hybrid model that integrates hydrodynamic constraints with a feed-forward neural network to predict the fuel consumption and speed of a container vessel. They compared and reported the advantage of using a hybrid model over a theoretical-only model for fuel consumption predictions during validation in extreme environmental conditions, although it is noted that their theoretical model does not include the effect of added resistance in waves. In addition, the theoretical model seems to be superior to the range of operating values, which suggests that its performance could have been improved in the more extreme environmental conditions if wave data and a theoretical wave model were included. The data were collected over a narrow vessel speed variance which may have limited the network training and have affected the comparisons between methods.

### 3. The Open-Water Propeller

The open-water propeller refers to a propeller working in uniform inflow, independent of the influence of the ship to which it may be fitted. Open-water tests allow taking measurements of thrust (T) and torque (Q) taken for a range of speed of advance ($V_A$) and propeller revolutions (n) of a propeller running in undisturbed water. The recorded thrust and torque are then nondimensionalized applying the relationships shown in Equations (1) and (2).

$$K_T = \frac{T}{\rho \cdot n^2 \cdot D^4} \tag{1}$$

$$K_Q = \frac{Q}{\rho \cdot n^2 \cdot D^5} \tag{2}$$

where D is the diameter of the propeller and $\rho$ is the mass density of the water. The open-water performance of the propeller can be computed using Equation (3).

$$\eta_o = \frac{J \cdot K_T}{2 \cdot \pi \cdot K_Q} \tag{3}$$

where J is the advance ratio, defined as follows:

$$J = \frac{V_A}{n \cdot D} \tag{4}$$

Now, let us define $K_{To}$ as the zero-speed thrust coefficient or the thrust coefficient $K_T$ when the value of the propeller advance ratio J is zero ($K_{To} = K_T(J = 0)$), and $K_{Qo}$ as the zero-speed torque coefficient or the torque coefficient $K_Q$ when the value of the propeller advance ratio J is zero ($K_{Qo} = K_Q(J = 0)$).

We can further define the coefficient $J_{ot}$ as a zero-thrust propeller advance ratio or the propeller advance ratio J such that the thrust developed by the propeller is zero ($J_{ot} = J(K_T = 0)$), and $J_{oq}$ as the zero-torque propeller advance ratio or the propeller advance ratio J such that the torque delivered to the propeller is zero ($J_{oq} = J(K_Q = 0)$).

Since $K_Q$ is assumed to be a smooth continuous derivable curve connecting the points ($J_{oq}$, 0) and (0, $K_{Qo}$), let us express the first derivative of $K_Q$ with respect to J as

$$\frac{dK_Q}{dJ} = -\frac{K_{Qo}}{f\left(J_{oq}\right)} \frac{d}{dJ} f(J) = \frac{K_Q - K_{Qo}}{f(J)} \cdot \frac{d}{dJ} f(J) \tag{5}$$

from where it seems to be possible to express the first derivative of $K_Q$ as a function of $K_Q$. The simplest approximation of $\frac{dK_Q}{dJ}$ could be approximated by constant value $c_1$,

$$\frac{dK_Q}{dJ} \approx c_1 \tag{6}$$

Solving for $K_Q$ by integrating Equation (6) would lead to expressing $K_Q$ as a straight line. Thus, it follows that the simplest non-trivial approximation for $\frac{dK_Q}{dJ}$ could be that of linear dependency with $K_Q$:

$$\frac{dK_Q}{dJ} \approx c_1 + c_2 \cdot K_Q \tag{7}$$

Solving for $K_Q$ by integrating Equation (7) yields:

$$K_Q = c_3 \cdot e^{c_2 \cdot J} - \frac{c_1}{c_2} \tag{8}$$

$c_1$, $c_2$ and $c_3$ are constants. Then, (1) $K_{Qo} = c_3 - \frac{c_1}{c_2}$; (2) $J_{oq} = \frac{1}{c_2} \cdot \ln\left(\frac{c_1}{c_2 \cdot c_3}\right)$; and (3) $k_q = c_2$, yields the following expression for $K_Q$:

$$K_Q = K_{Qo} \cdot \left(1 - \frac{e^{k_q \cdot J} - 1}{e^{k_q \cdot J_{oq}} - 1}\right) \tag{9}$$

Similarly, the thrust coefficient ($K_T$) can be represented by the following expression:

$$K_T = K_{To} \cdot \left(1 - \frac{e^{k_t \cdot J} - 1}{e^{k_t \cdot J_{ot}} - 1}\right) \tag{10}$$

Given Equations (9) and (10), the efficiency of the open-water propeller ($\eta_o$) can be expressed as

$$\eta_o = \frac{J}{2\pi} \cdot \frac{K_{To}}{K_{Qo}} \cdot \frac{\left(e^{k_q \cdot J_{oq}} - 1\right)}{\left(e^{k_t \cdot J_{ot}} - 1\right)} \cdot \frac{\left(e^{k_t \cdot J_{ot}} - e^{k_t \cdot J}\right)}{\left(e^{k_q \cdot J_{oq}} - e^{k_q \cdot J}\right)} \tag{11}$$

A few examples illustrate the applicability of Equations (9) and (10). Figures A1–A5 in the appendix show the open-water characteristics of a few propellers used to test several Series 60 Models [50]. Figure A6 shows the open-water characteristics of the propeller KP458 [51], used to test the KRISO Very Large Crude-oil Carrier 2 (KVLCC2) [52]. Regressing Equations (9) and (10) to the open-water propeller data J, $K_T$ and $K_Q$ leads to the fitting parameters $K_{To}$, $J_{ot}$, $k_t$, $K_{Qo}$, $J_{oq}$, and $k_q$, achieving the goodness-of-fit characterized by the determination coefficients $R^2(K_T)$, $R^2(K_Q)$, shown in Tables A1–A6. The curves in Figures A1–A6 were created with Equations (9)–(11) and the fitting parameters $K_{To}$, $J_{ot}$, $k_t$, $K_{Qo}$, $J_{oq}$, and $k_q$, shown in each corresponding table.

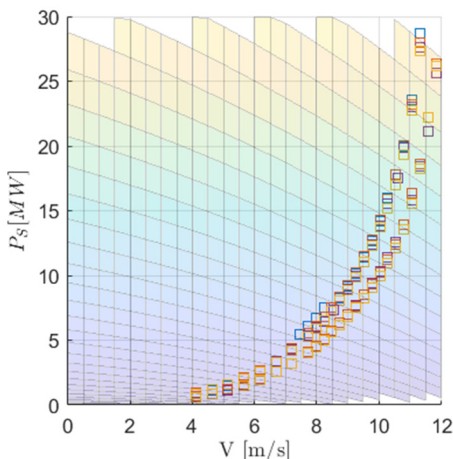 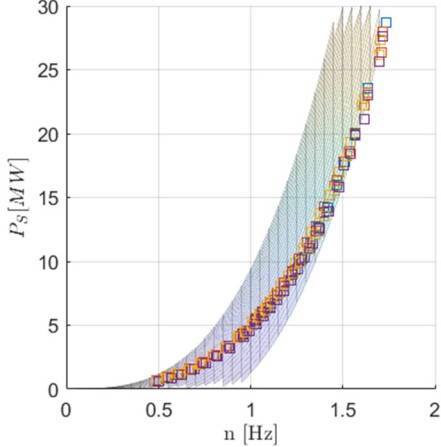

**Figure 1.** *Cont.*

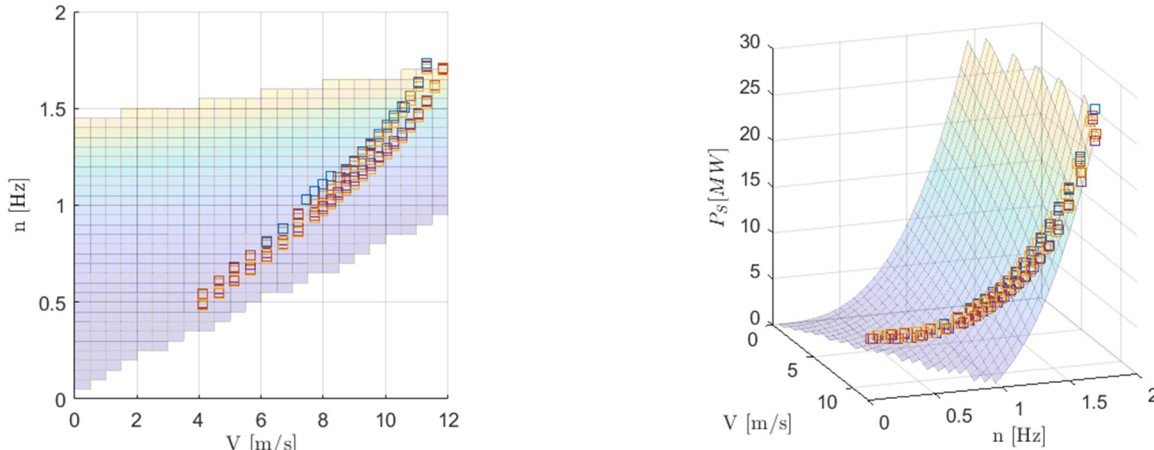

**Figure 1.** Shaft Power ($P_S$) from Equation (23) and Series 60 Models 4221, 4280, 4281, 4282 data.

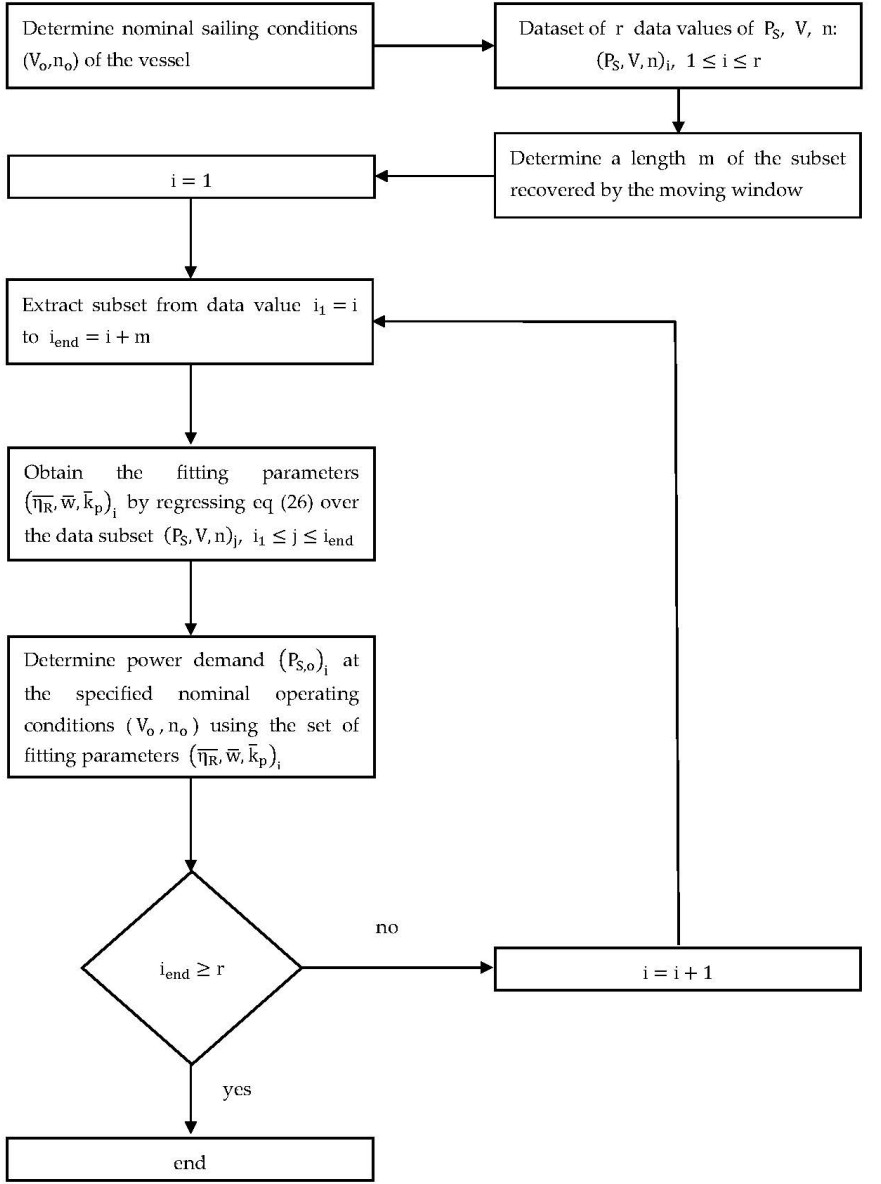

**Figure 2.** Flowchart diagram of the proposed process to obtain the time evolution of the performance of the vessel.

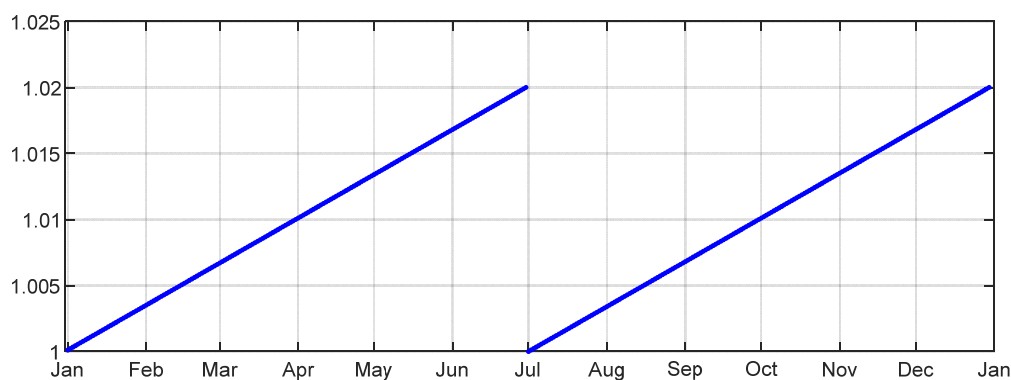

**Figure 3.** Biofouling coefficient.

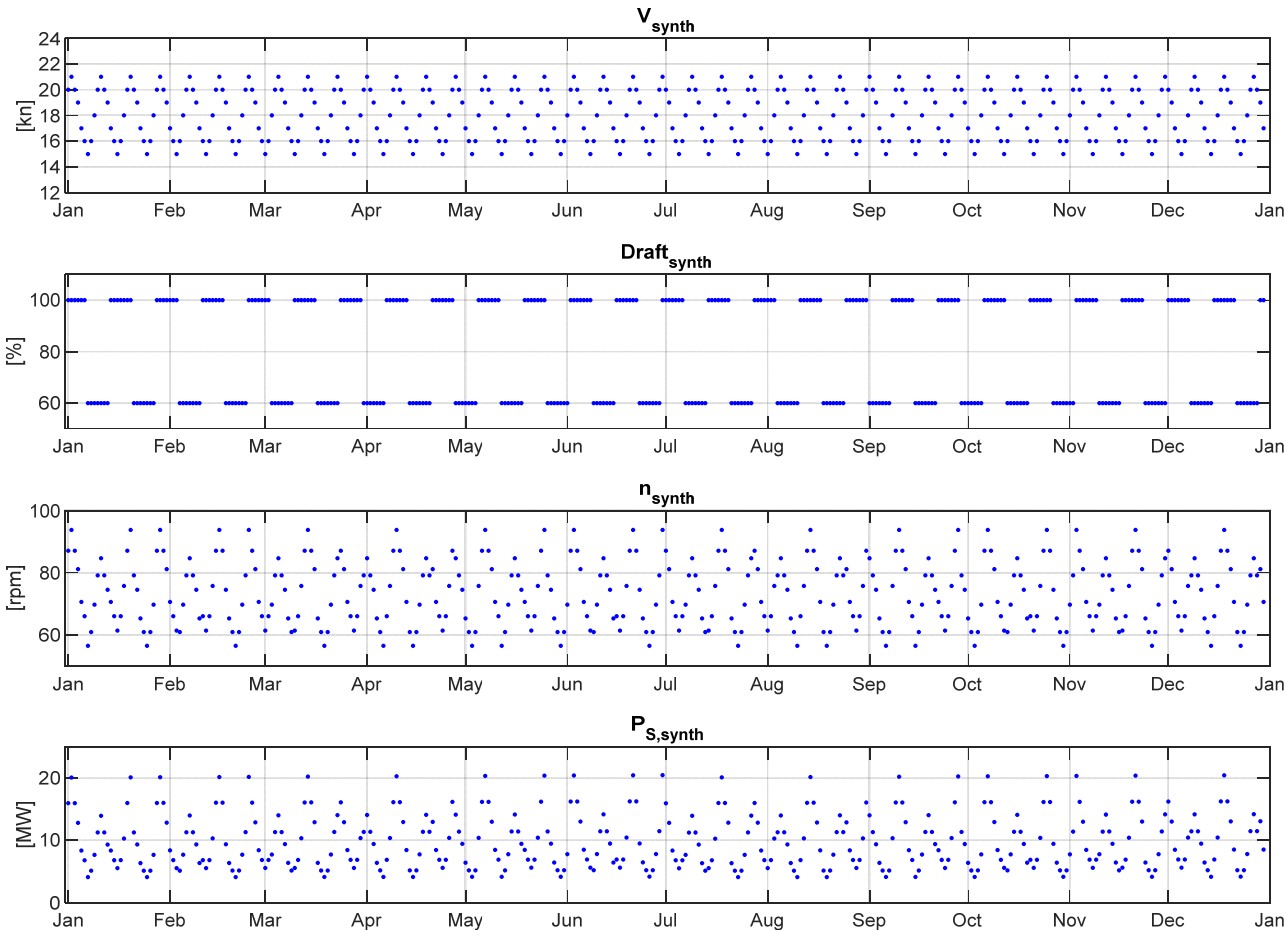

**Figure 4.** Synthetic dataset.

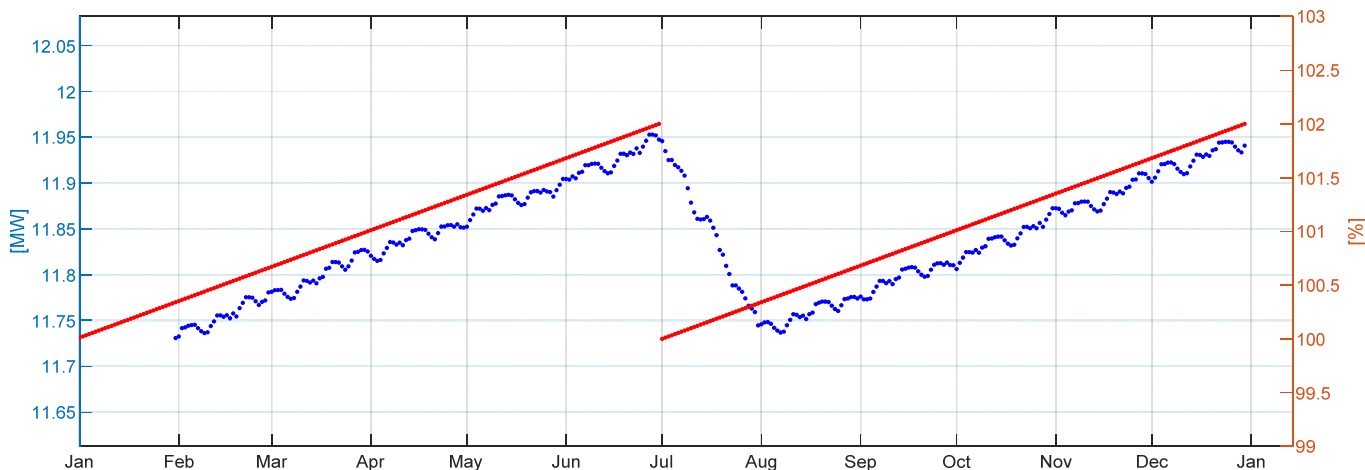

**Figure 5.** First 30-day window dataset.

**Figure 6.** Time evolution of vessel performance.

## 4. The Full-Scale Vessel

The effect of moving the propeller from an open-water scenario to a behind-the-hull scenario is typically quantified through the inclusion of the wake fraction (w), the thrust deduction coefficient (t), and the rotative relative efficiency ($\eta_R$).

The wake fraction (w) accounts for the loss of speed of the water due to the presence of the hull. The wake is the combination of the boundary layer associated with skin friction,

the flow velocities occasioned by the streamlined form of the ship and the orbital velocities of the waves created by the ship. If the ship speed is V and the average velocity of the water relative to the hull at the propeller position is $V_A$, the wake speed, $V - V_A$, leads to the definition of the non-dimensional wake fraction as $w = 1 - V_A/V$.

The action of the propeller causes the water in front of it to be sucked towards the propeller. This results in extra resistance on the hull. The thrust force (T) on the propeller must overcome both the ship's towing resistance ($R_T$) and the extra resistance on the hull due to the sucking action of the propeller. The difference between the thrust force (T) and the towing resistance ($R_T$), $T - R_T$, corresponds to a loss of thrust. Thus, a non-dimensional thrust deduction coefficient (t) can be defined as $t = 1 - R_T/T$.

Since water closes in around the stern, the flow through the propeller disc will not be the same everywhere and will not, in general, be parallel to the shaft line. These effects can be combined and expressed as a relative rotative efficiency ($\eta_R$) as $\eta_R = \eta_B/\eta_o$, where $\eta_B$ is the behind-the-hull propeller efficiency and $\eta_o$ is the open-water propeller efficiency.

The power measured in the shaft is the shaft power ($P_S$) delivered to the shafting system by the propelling machinery ($P_S = P_D/\eta_S$), where the shafting efficiency ($\eta_s$) is a measure of the power lost in shaft bearings and a stern tube. Thus, shaft power ($P_S$) and towing resistance ($R_T$) can be expressed as shown in Equations (12) and (13) respectively,

$$P_S = \frac{1}{\eta_S \cdot \eta_R} \cdot 2 \cdot \pi \cdot \rho \cdot n^3 \cdot D^5 \cdot K_{QS}(J, Re) \tag{12}$$

$$R_T = (1 - t) \cdot \rho \cdot n^2 \cdot D^4 \cdot K_{TS}(J, Re) \tag{13}$$

where $K_{QS}$ and $K_{TS}$ are the characteristics of the full-scale propeller, following [10], calculated from the open-water characteristics of the scaled model propeller, $K_{QM}$ and $K_{TM}$, as follows,

$$K_{QS}(J, Re) = (K_{QM} - \Delta K_Q) \tag{14}$$

$$K_{TS}(J, Re) = (K_{TM} - \Delta K_T) \tag{15}$$

where

$$\Delta K_Q = 0.25 \cdot \frac{c}{D} \cdot Z \cdot \Delta C_D \tag{16}$$

$$\Delta K_T = -0.3 \cdot \frac{P}{D} \cdot \frac{c}{D} \cdot Z \cdot \Delta C_D \tag{17}$$

The difference in drag coefficient, $\Delta C_D$, is

$$\Delta C_D = C_{DM} - C_{DS} \tag{18}$$

where

$$C_{DM} = 2 \cdot \left(1 + 2 \cdot \frac{t}{c}\right) \cdot \left[\frac{0.044}{(Re_{0.7})^{1/6}} - \frac{5}{(Re_{c0})^{2/3}}\right] \tag{19}$$

and

$$C_{DS} = 2 \cdot \left(1 + 2 \cdot \frac{t}{c}\right) \cdot \left[1.89 + 1.62 \cdot \log \frac{c}{k_P}\right] \tag{20}$$

thus,

$$\Delta C_D = 3.78 \cdot \left(1 + 2 \cdot \frac{t}{c}\right) \cdot \left[\frac{0.0233}{(Re_{0.7})^{1/6}} - \frac{2.6455}{(Re_{c0})^{2/3}} - 0.8571 \cdot \log \frac{c}{k_P} - 1\right] \tag{21}$$

where Z is the number of propeller blades, c is the chord length, t is the maximum thickness, P/D is the pitch ratio and $Re_{07}$ is the local Reynolds number with Kempf's definition at the open-water test,

$$Re_{0.7} = \frac{c_{0.7} \cdot \sqrt{V_A^2 + (0.7 \cdot \pi \cdot n \cdot D)^2}}{\nu} = \frac{c_{0.7} \cdot n \cdot D}{\nu} \cdot \sqrt{J^2 + (0.7 \cdot \pi)^2} \tag{22}$$

They are defined for the representative blade section, such as at r/R = 0.7. $k_p$ denotes the blade roughness, the standard value of which is set $k_p = 30 \cdot 10^{-6}$ m. Either $Re_{07}$ or $Re_{c0}$ must not be lower than $2 \cdot 10^5$.

Given Equations (9), (12) and (14) providing mathematical expressions for $P_S$ and $K_Q$, it follows:

$$P_S = \frac{1}{\eta_s \cdot \eta_R} \cdot 2 \cdot \pi \cdot \rho \cdot n^3 \cdot D^5 \cdot \left[ K_{Qo} \cdot \left( \frac{e^{k_q \cdot J_{oq}} - e^{k_q \cdot J}}{e^{k_q \cdot J_{oq}} - 1} \right) - \Delta K_Q \right] \tag{23}$$

where the advance ratio, $J = (1 - w) \cdot V/n \cdot D$. Figure 1 shows the shaft power surface created by Equation (23) if assumed w = 0.319 and $\eta_R = 1.018$. The towing resistance, from Equations (10), (13) and (15),

$$R_T = (1 - t) \cdot \rho \cdot D^4 \cdot n^2 \cdot \left[ K_{To} \cdot \left( \frac{e^{k_t \cdot J_{ot}} - e^{k_t \cdot J}}{e^{k_t \cdot J_{ot}} - 1} \right) - \Delta K_T \right] \tag{24}$$

which in combination with Equation (23) allows expressing the shaft power demand $P_S$ as a closed-form expression in the form

$$P_S = P_S \left( \rho, \nu, K_{To}, J_{ot}, k_t, K_{Qo}, J_{oq}, k_q, k_p, \frac{P_{0.7}}{D}, \frac{t_{0.7}}{D}, \frac{c_{0.7}}{D}, Z, \eta_S, \eta_R, t, n, D, R_T \right) \tag{25}$$

Regressing Equation (25) over a selection of Series 60 Models and the KVLCC2 data, using parameters $K_{To}$, $J_{ot}$, $k_t$, $K_{Qo}$, $J_{oq}$, and $k_q$ listed in Tables A1–A6, leads to the goodness-of-fit characterized by the determination coefficients $R^2(P_S)$ shown in Table 1.

**Table 1.** Goodness-of-fit achieved by the model in each case tested.

| Model | Propeller | $R^2(P_S)$ |
|---|---|---|
| 4210 | 3378 | 0.999868 |
| 4213 | 3379 | 0.999692 |
| 4214 | 3377 | 0.999994 |
| 4215 | 3378 | 0.999971 |
| 4218 | 3380 | 0.999655 |
| 4221 | 3376 | 0.999626 |
| 4256 | 3380 | 0.999651 |
| 4260 | 3377 | 0.999994 |
| 4272 | 3378 | 0.999919 |
| 4280 | 3376 | 0.999326 |
| 4281 | 3376 | 0.999254 |
| 4282 | 3376 | 0.999674 |
| KVLCC2 | KP458 | 0.999865 |

The Series 60 Models followed the speed and power prediction method detailed in [53], in which propeller full scaling was not applied, thus $\Delta K_T = \Delta K_Q = 0$. For the KVLCC2 propeller full scaling, it was assumed that $\rho = 1025.9$, $\nu = 1.18831 \cdot 10^{-6}$, $P_{0.7}/D = 0.7212$, $t_{0.7}/D = 15.6 \cdot 10^{-3}$, $c_{0.7}/D = 0.2338$, and $Re_{c0} = 2 \cdot 10^5$.

## 5. Performance Evaluation

Vessel performance evaluation tries to quantify the speed reduction or increase of the power demand that results from the in-service degradation of the vessel.

In a general sense, it can be assumed that the wake fraction depends on the vessel sailing conditions. It also makes sense that the progressive increase of frictional resistance due to the biofouling growth in the hull must have some effect in the set of all the possible values of the wake fraction. Should this effect happen uniformly over the whole set of possible values of w, then the time evolution of the average wake fraction ($\overline{w}$) must capture the increase of hull frictional resistance over time.

In other words, it is expected that the average of all the possible wake fraction values of a smooth hull sailing under all possible sailing conditions to be smaller than the average of all possible wake fraction values of an otherwise same hull but with a significantly higher level of roughness.

Similar reasoning can be applied to the relative rotative efficiency ($\eta_R$) and the blade roughness ($k_p$), where the time evolution of the average of the relative rotative efficiency ($\overline{\eta_R}$) and blade roughness ($\overline{k_p}$), could be seen as a manifestation of the variability over time of the propeller efficiency range.

The patent application "Obtaining and Utilizing Power Demand data of Self-Propelled Vehicles. (U.S. Patent Application no. 17/225,019) U.S. Patent and Trademark Office. 2021" describes the following method to estimate the evolution of the performance of a vessel over time: given a time series of an operational vessel sailing data $(P_S, V, n)_i$, $1 \le i \le r$, a series of values $\left(\overline{\eta_R}, \overline{w}, \overline{k_p}\right)_i$ can be obtained by iteratively applying regression analysis of Equation (23) over the data subset extracted by a moving window along with the time series dataset. The evolution of values $\left(\overline{\eta_R}, \overline{w}, \overline{k_p}\right)_i$ then captures the average time degradation of hull and propeller.

$$\hat{P_S} = \frac{1}{\eta_s \cdot \overline{\eta_R}} \cdot 2 \cdot \pi \cdot \rho \cdot n^3 \cdot D^5 \cdot K_{Qo} \cdot \left( \frac{e^{k_q \cdot J_{oq}} - e^{k_q \cdot \frac{(1-\overline{w}) \cdot V}{n \cdot D}}}{e^{k_q \cdot J_{oq}} - 1} - \Delta K_Q\left(V, n, \overline{w}, \overline{k_p}\right) \right) \tag{26}$$

If pre-defined nominal conditions $(n_o, V_o)$ are chosen, the calculation of the vessel shaft power demand $(P_{S,o})$ at $(n_o, V_o)$ for each $\left(\overline{\eta_R}, \overline{w}, \overline{k_p}\right)_i$ would yield the evolution over time of the vessel shaft power demand as if the vessel would have continuously sailed at the nominal conditions $(n_o, V_o)$. Thus, the series values $(P_{S,o})_i$, $1 \le i \le r$,

$$P_{S,o} = \frac{1}{\eta_s \cdot \overline{\eta_R}} \cdot 2 \cdot \pi \cdot \rho \cdot n_o^3 \cdot D^5 \cdot K_{Qo} \cdot \left( \frac{e^{k_q \cdot J_{oq}} - e^{k_q \cdot \frac{(1-\overline{w}) \cdot V_o}{n_o \cdot D}}}{e^{k_q \cdot J_{oq}} - 1} - \Delta K_Q\left(V_o, n_o, \overline{w}, \overline{k_p}\right) \right) \tag{27}$$

reflect the time evolution of the performance of the vessel. Figure 2 shows a flowchart describing this process.

As a simple but illustrative example of the application of this method, let us consider a synthetic dataset of 365 data points (each point corresponding to a day of the year). A variable "DN", as "DayNumber", can be defined as an incremental counter between 1 and 365. For this example, it will be assumed that the vessel speed, expressed in knots, changes daily following the equation:

$$V_{synth} = \left\lfloor 18.5 + \left( \frac{7}{\pi} \cdot \arcsin\left( \sin\left( \frac{20 \cdot \pi}{9} \cdot DN \right) \right) \right) \right\rfloor \tag{28}$$

where the symbols $\lfloor \cdot \rfloor$ indicate the floor function, i.e., the function that takes as input a real number, and gives as output the greatest integer less than or equal to that number. he draft, in percent displacement, changes weekly following the equation:

$$Draft_{synth} = 60 + 40 \cdot \left( WN - 2 \left\lfloor \frac{WN}{2} \right\rfloor \right) \tag{29}$$

where "WN" is the "WeekNumber" calculated as

$$WN = 1 + \left[\frac{DN}{7}\right] \tag{30}$$

As noted by the following Table 2, Equation (30) provides two values, namely 60 when the "WeekNumber" is an even number, and 100 when the "WeekNumber" is an odd number:

**Table 2.** Synthetic data draft pattern.

| WN | $WN - 2[\frac{WN}{2}]$ | $60 + 40 \cdot (WN - 2[\frac{WN}{2}])$ |
|----|------------------------|----------------------------------------|
| 1 | 1 | 100 |
| 2 | 0 | 60 |
| 3 | 1 | 100 |
| 4 | 0 | 60 |
| ⋮ | ⋮ | ⋮ |

The values of propeller revolutions, wake fraction, relative rotative efficiency and shaft power demand are obtained by matching the synthetic speed and draft values obtained with Equations (28) and (30) to the Series 60 Model 4280 data.

Then, let us simulate the progressive increase of power demand due to biofouling by a coefficient that starts at a value of 1 and linearly increases over time. An event such as a hull cleaning, propeller polishing, or the application of a new coating during dry dock, is simulated by dropping the biofouling coefficient back to a value of 1, before increasing once more over time.

If the vessel begins operation on 1 January having a biofouling coefficient with a value of 1 and rises to a value of 1.02 by 30 June, the vessel can be cleaned at the half-year mark on July 1 such that the biofouling coefficient drops back down to 1. Figure 3 shows the time evolution of the biofouling coefficient.

The synthetic shaft power demand considered during this simulation is the result of multiplying the biofouling coefficient by the shaft power demand from the Series 60 Model 4280 obtained by matching the obtained synthetic speed and draft with the Series 60 Model 4280 published data.

Figure 4 shows the resulting vessel synthetic speed ($V_{synth}$), synthetic draft ($Draft_{synth}$), synthetic rate of propeller rotation ($n_{synth}$), and synthetic shaft power demand ($P_{S,synth}$) with the biofouling coefficient already applied.

We can now specify the vessel nominal conditions as $V_o = 20$ knots $= 10.288$ m/s and $n_o = 80$ rpm $= 1.333$ Hz. In addition, the size of the moving window is defined as 30 days.

The first step is to take data within the first 30 days. Figure 5 shows the first 30-day window extracted from the synthetic dataset, between 1 January and 30 January.

Then, using regression analysis of Equation (26) over the 30-day data selection shown in Figure 5 would be possible to obtain the values $\left(\overline{w}, \overline{\eta_R}, \overline{k_P}\right)_1$, and using Equation (27) would yield $(P_{S,o})_1$.

The 30-day moving window can then be advanced by 1 day, fitting Equation (26) the data subset from 2 January to 31 January will provide new values $\left(\overline{w}, \overline{\eta_R}, \overline{k_P}\right)_2$, used again to estimate the shaft power demand at $(V_o, n_o)$ will yield $(P_{S,o})_2$, and so on.

The process finalizes when the 30-day moving window arrives at the end of the dataset. Then, the series of values $(P_{S,o})_1, (P_{S,o})_2, \ldots, (P_{S,o})_r$ can be interpreted as the shaft power demand that would have been obtained if the vessel would have continuously sailed at the fixed conditions of $V_o = 20$ knots and $n_o = 80$ rpm.

Figure 6 shows the series of values $(P_{S,o})_1, (P_{S,o})_2, \ldots, (P_{S,o})_r$ obtained in the example, referenced to the left axis; as well as the previously defined biofouling coefficient, referenced to the right axis. As it can be seen in the figure, the series of values $(P_{S,o})_1, (P_{S,o})_2, \ldots, (P_{S,o})_r$ reflect changes in the in-service degradation of the hull. They successfully identified

the hull cleaning event that occurred on July 1, the maximum increase of shaft power demand, and the hull degradation rate.

## 6. Conclusions

The prediction of the power demand of a self-propelled full-scale vessel has been an intensive area of research in Naval Architecture for more than a hundred years. Recent changes in the regulatory framework and increase of fuel oil prices incentivized the industry to seek a high accuracy level on model predictions of the operational power demand of vessels sailing under arbitrary speed, draft, trim, or weather conditions.

Current well-established models in Naval Architecture work well in the controlled environment of a Towing Tank Test but have not been successfully extended over vessel operational sailing data.

Recent development in Machine Learning models intensified the research on this topic producing a plethora of articles in the literature covering approaches with a progressive level of sophistication. However, statistics and machine learning models make it difficult to understand the actual relationship between parameters. In addition, they require the dataset to be an unbiased sample, and this seldom happens because operational constraints produce preferred speeds, drafts, and trims in vessel operational sailing data.

The motivation of the research outlined in this paper is the derivation of a mathematical model both consistent with Principles of Naval Architecture and with sounding applicability over operational sailing vessel data.

This paper derives minimal inversible mathematical expressions for the torque ($K_Q$) and thrust ($K_T$) open-water characteristics of the scaled model propeller, from where it is possible to arrive at the following set of equations,

$$\frac{P_S \cdot \eta_s \cdot \eta_R}{2 \cdot \pi \cdot \rho \cdot n^3 \cdot D^5} = K_{Qo} \cdot \left( \frac{e^{k_q \cdot J_{oq}} - e^{k_q \cdot J}}{e^{k_q \cdot J_{oq}} - 1} \right) - \Delta C_D \cdot 0.25 \cdot \frac{c_{0.7}}{D} \cdot Z \tag{31}$$

$$\frac{R_T}{(1-t) \cdot \rho \cdot D^4 \cdot n^2} = K_{To} \cdot \left( \frac{e^{k_t \cdot J_{ot}} - e^{k_t \cdot J}}{e^{k_t \cdot J_{ot}} - 1} \right) + \Delta C_D \cdot 0.3 \cdot \frac{P_{0.7}}{D} \cdot \frac{c_{0.7}}{D} \cdot Z \tag{32}$$

$$\Delta C_D = 3.78 \cdot \left( 1 + 2 \cdot \frac{t_{0.7}}{c_{0.7}} \right) \left\{ \frac{0.0233}{(Re_{0.7})^{1/6}} - \frac{2.6455}{(Re_{c0})^{2/3}} - 0.8571 \cdot \log \frac{c_{0.7}}{k_p} - 1 \right\} \tag{33}$$

$$Re_{0.7} = \frac{c_{0.7}}{\nu} \cdot n \cdot D \cdot \sqrt{J^2 + (0.7 \cdot \pi)^2} \tag{34}$$

which allow a closed-form characterization of the shaft power demand,

$$P_S = P_S \left( \rho, \nu, D, K_{To}, J_{ot}, k_t, K_{Qo}, J_{oq}, k_q, \frac{P_{0.7}}{D}, \frac{t_{0.7}}{D}, \frac{c_{0.7}}{D}, Z, \eta_S, \eta_R, t, k_p, n, R_T \right) \tag{35}$$

where,

- $\rho$ is the mass density of the water;
- $\nu$ is the kinematic viscosity of the water;
- D is the propeller diameter;
- $K_{Qo}$, $J_{oq}$, $k_q$, $K_{To}$, $J_{ot}$ and $k_t$ are the open-water characteristics of the scaled model propeller;
- $P_{0.7}/D$ is the pitch ratio at the blade section $r/R = 0.7$;
- $t_{0.7}$ is the propeller maximum blade thickness at the blade section $r/R = 0.7$;
- $c_{0.7}$ is the propeller blade chord length at the blade section $r/R = 0.7$;
- Z is the number of propeller blades;
- $\eta_S$ is the shaft efficiency;
- $\eta_R$ is the relative rotative efficiency;
- t is the thrust deduction fraction;

- $k_p$ is the blade roughness;
- n are the propeller revolutions;
- $R_T$ is the vessel towing resistance.

The validation of the model represented by Equations (31)–(34) was conducted with published data of a few Series 60 models and the KRISO Very Large Crude-oil Carrier 2 (KVLCC2). In all cases, the model explained more than 99.9% of the data variability.

It is noteworthy to mention that the extension to the equations derived for the open-water scaled model propeller to the full-scale vessel was performed following the Naval Architecture well-accepted body of knowledge. The integration of the wake fraction, thrust deduction fraction, and relative rotative efficiency in the open-water scaled model equations was performed following the definition of these parameters. Thus, as long as the equations derived for the thrust and torque coefficients appropriately explain the open-water characteristics of the scaled model propeller, by definition of w, t and $\eta_R$, Equation (35) must be an accurate representation of the inner consistency of the variables involved.

This paper also prescribes a practical method for measuring changes in hull and propeller performance. To ensure reproducibility, an example over a synthetic dataset was provided.

Present and future work focuses on three areas of interest:

Independent characterization of the parameters t, $\eta_R$, $k_P$, $R_T$; the direct application of Equations (31)–(34) to each datapoint ($P_S$, n, V) produces an undetermined system, from where more constraints are needed.; connection of the derived mathematical expressions for $K_T$ and $K_Q$ with well-accepted propeller theory.

**Funding:** This research received no external funding.

**Conflicts of Interest:** The author declares no conflict of interest.

## Appendix A

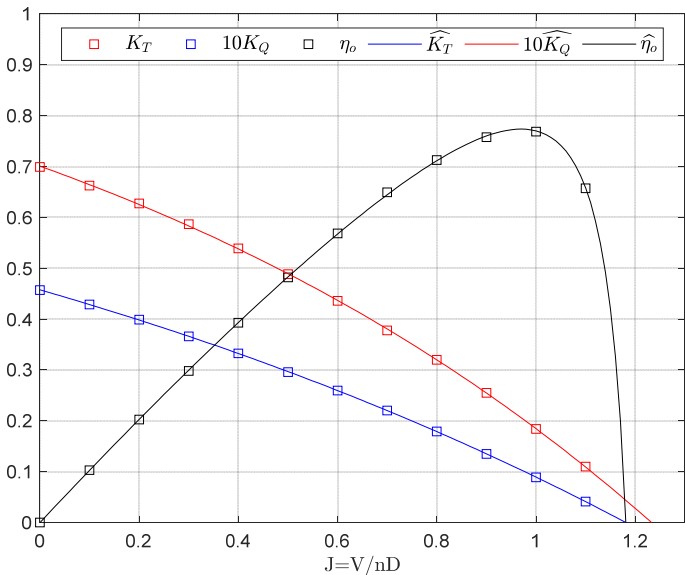

**Figure A1.** Propeller DTMB 3376 open-water characteristics. Data from [54].

**Table A1.** Fitting parameters and $R^2$ scores regressing Equations (9)–(11) to DTMB 3376 open-water propeller data.

| | |
|---|---|
| $K_{To}$ | 0.4575 |
| $J_{ot}$ | 1.181 |
| $k_t$ | 0.5122 |
| $K_{Qo}$ | 0.0701 |
| $J_{oq}$ | 1.2327 |
| $k_q$ | 0.7298 |
| $R^2(K_T)$ | 0.999987 |
| $R^2(K_Q)$ | 0.999920 |
| $R^2(\eta_o)$ | 0.999949 |

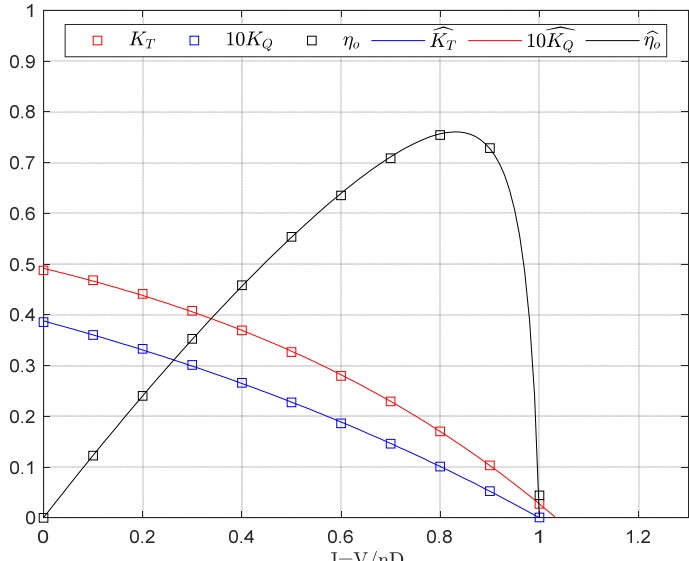

**Figure A2.** Propeller DTMB 3377 open-water characteristics. Data from [54,55].

**Table A2.** Fitting parameters and $R^2$ scores regressing Equations (9)–(11) to DTMB 3377 open-water propeller data.

| | |
|---|---|
| $K_{To}$ | 0.3880 |
| $J_{ot}$ | 1.0000 |
| $k_t$ | 0.7115 |
| $K_{Qo}$ | 0.0492 |
| $J_{oq}$ | 1.0331 |
| $k_q$ | 0.2231 |
| $R^2(K_T)$ | 0.999844 |
| $R^2(K_Q)$ | 0.999824 |
| $R^2(\eta_o)$ | 0.997424 |

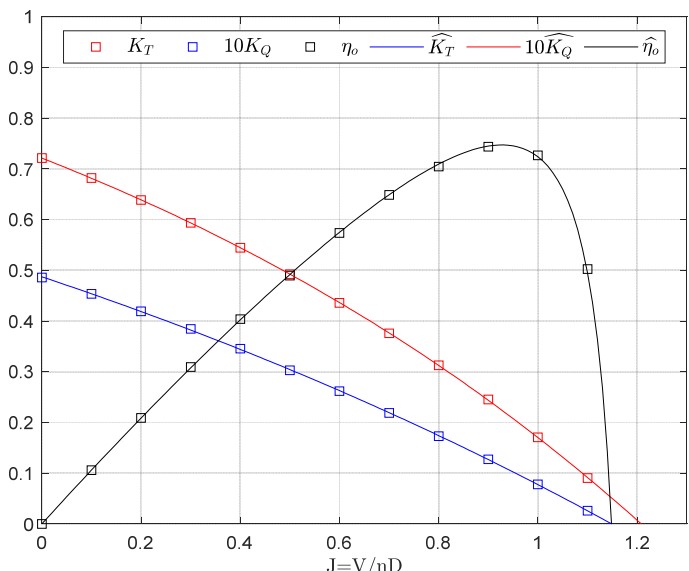

**Figure A3.** Propeller DTMB 3378 open-water characteristics. Data from [54].

**Table A3.** Fitting parameters and $R^2$ scores regressing Equations (9)–(11) to DTMB 3378 open-water propeller data.

| | |
|---|---|
| $K_{To}$ | 0.4872 |
| $J_{ot}$ | 1.1486 |
| $k_t$ | 0.4307 |
| $K_{Qo}$ | 0.0721 |
| $J_{oq}$ | 1.2082 |
| $k_q$ | 0.6835 |
| $R^2(K_T)$ | 0.999947 |
| $R^2(K_Q)$ | 0.999999 |
| $R^2(\eta_o)$ | 0.999854 |

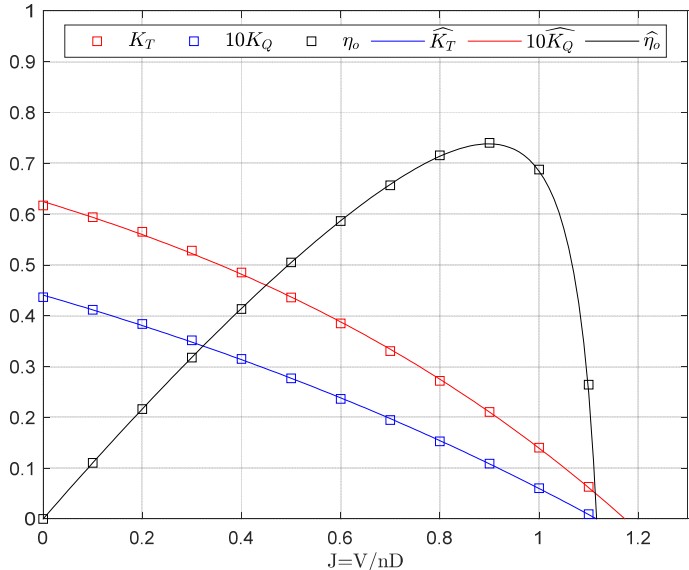

**Figure A4.** Propeller DTMB 3379 open-water characteristics. Data from [54,56].

**Table A4.** Fitting parameters and $R^2$ scores regressing Equations (9)–(11) to DTMB 3379 open-water propeller data.

| | |
|---|---|
| $K_{To}$ | 0.4407 |
| $J_{ot}$ | 1.1165 |
| $k_t$ | 0.5654 |
| $K_{Qo}$ | 0.0625 |
| $J_{oq}$ | 1.1732 |
| $k_q$ | 0.9210 |
| $R^2(K_T)$ | 0.999746 |
| $R^2(K_Q)$ | 0.999544 |
| $R^2(\eta_o)$ | 0.999566 |

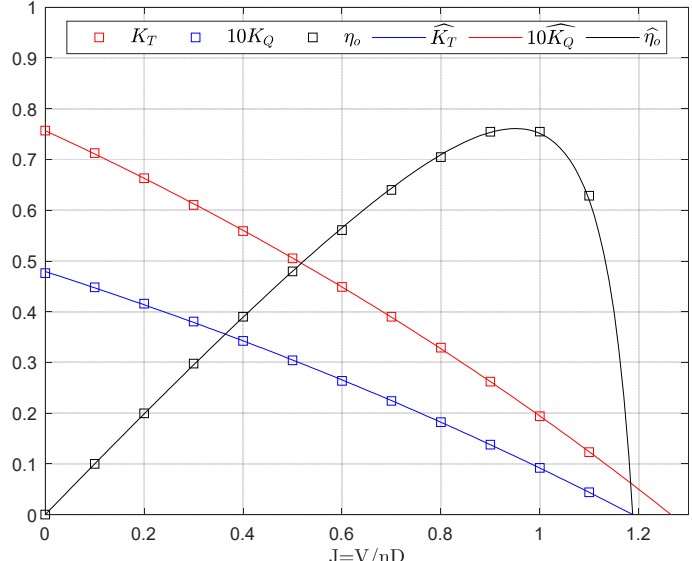

**Figure A5.** Propeller DTMB 3380 open-water characteristics. Data from [54].

**Table A5.** Fitting parameters and $R^2$ scores regressing Equations (9)–(11) to DTMB 3380 open-water propeller data.

| | |
|---|---|
| $K_{To}$ | 0.4791 |
| $J_{ot}$ | 1.1879 |
| $k_t$ | 0.4017 |
| $K_{Qo}$ | 0.0757 |
| $J_{oq}$ | 1.2654 |
| $k_q$ | 0.4292 |
| $R^2(K_T)$ | 0.999892 |
| $R^2(K_Q)$ | 0.999967 |
| $R^2(\eta_o)$ | 0.999796 |

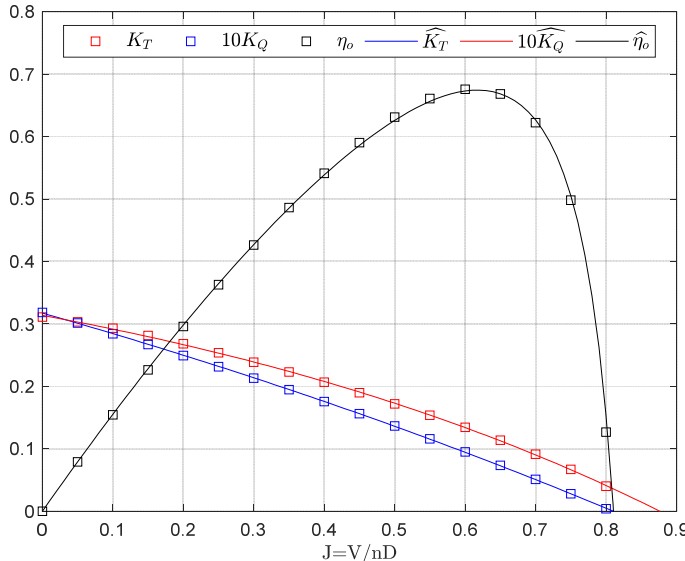

**Figure A6.** Propeller KP458 open-water characteristics. Data from [51].

**Table A6.** Fitting parameters and $R^2$ scores regressing Equations (9)–(11) to KP458 open-water propeller data.

| | |
|---|---|
| $K_{To}$ | 0.3174 |
| $J_{ot}$ | 0.8107 |
| $k_t$ | 0.4702 |
| $K_{Qo}$ | 0.0314 |
| $J_{oq}$ | 0.8765 |
| $k_q$ | 1.1326 |
| $R^2(K_T)$ | 0.999969 |
| $R^2(K_Q)$ | 0.999787 |
| $R^2(\eta_o)$ | 0.998864 |

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
