# Peer review of "A Power Demand Analytical Model of Self-Propelled Vessels"

_jmse, doi:10.3390/jmse9121450_

Round 1

Reviewer 1 Report

The last section on performance evaluation , a patented approach, is not clear. The researchers need explicit information to check the repeatability of the results presented in the last section. 

The authors have a new representation of propeller open water characteristics. This is a fine addition to propeller algorithms. The examples provided are fine. 

With some review and explicit definition of the performance calculations it will be a fine contribution.

Author Response

The response to the reviewer is in the file attached

Reviewer 2 Report

Dear Author,

thanks for this interesting paper that investigate the definition of closed-form mathematical expressions that allow expressing the shaft power demand of a vessel as a function of a defined set of input parameters (i.e. open-water characteristics of a propeller, towing ship resistance, propeller revolutions, propeller diameter, fluid density, thrust deduction fraction, shaft efficiency and relative rotative efficiency).

The paper investigates a relevant topic for the maritime world, especially in the last years with the constantly increased interest in the consumption monitoring and reduction due to the new regulations policies, the increase in the environmental issue sensibility, and the fuel prices. However, the paper has serious lacks and need to be strongly improved.

As a general comment, I would suggest checking the abstract that sounds not so clear, specifically in the input parameters for the developed model.

Follows other major comments:

  1. Introduction

The literature overview could be extended, for instance, I would suggest including the reference to the new EEXI/CII regulations (IMO Res MEPC. 334(76) – 2021).

  1. State of the art of vessel performance modeling

In this paragraph, I would recommend including more papers related to ship performance monitoring, for instance, Bocchetti et al. A Statistical Approach to Ship Fuel Consumption Monitoring, Journal of Ship Research, vol. 59, 2015

From Figure 1 to Figure 6 and from Table 1 to Table 6.

My suggestion is to move these figures and tables to a dedicated appendix. In this way, the paper will be more compact.

In the abovementioned figures, furthermore, is not clear which are the experimental data at which the curves are fitting, are the dots? Is not clearly understandable, please include this information in the graph's legend.

There is some missed information related to the Tables (from 1 to 6), specifically:

-  kt (the thrust curvature parameter) is not clearly explained in the text.

-  Please improve the representation of the table.

  1. The full-scale vessel

“Regressing the equation (20) over a selection of Series 60 Models and the KVLCC2”

Why consider several models of Series 60 and not extend the regression validation process to a different set of vessels, including other benchmark hulls (e.g. KCS or Regal Vessel)?

  1. Performance Evaluation

This paragraph represents the weak point of this study, why not consider real vessel performance data? As far as I remember, there is some data available in some papers about the one-year performance vessel dataset. However, if the Author want to continue to use a synthetic dataset, why not generate it whit a random or semi-random approach mapping some real (but not sharable) vessel performance dataset?

“ηÌ…Ì…RÌ… = 1.247” is a quite big number as rotative relative efficiency even if is an average value.

“The 30-day moving window can then be advanced by 1 day repeating the process in the data range January 2 – January 31, and so on. The process finalizes when the 30-day 306 moving window arrives at the end of the dataset. We can interpret the series of values PS,i as the shaft power demand that would have been obtained if the vessel would have continuously sailed at the fixed conditions of Vo = 20 knots and no = 80 rpm. Figure 11 shows the series of values PS,i obtained in the example, as well as the bio-fouling coefficient previously defined. The graph shows that the series of values PS,i reflect changes in the in-service degradation of the hull, successfully identifying the hull cleaning event that occurred on July 1, the maximum increase of shaft power demand, and the hull degradation rate.”

This part of the paragraph is not clear, please reshape/improve it.

  1. Conclusion

This paragraph is extremely short and doesn’t provide all the relevant information generally available in the conclusion paragraph such as the purpose of the investigation, strong points and limits of the models, and ongoing activities and future works.

Author Response

(The authors gave the same response as above.)

Round 2

Reviewer 2 Report

Dear Author, thanks for all the applied changes, the paper has been strongly improved, I have well accepted all the answers provided by the Author.

There is only one main major comment. In Par. 4 “Full Scale Vessel” the author explains what happens when the propeller is moved from the “open water” to the “behind the hull” condition. However, the effects of putting the propeller behind the hull are not related to the full-scale vessel (the same issue are in model scale between open water and propeller behind the hull model). Furthermore, there is no explanation about the full scaling procedures (seems that the propeller performance has not been corrected to take into account the effect of the full scaling process – please take a look at ITTC 7.5 – 02-03 – 01.4 “Performance, Propulsion 1978 ITTC Performance Prediction Method”, 2008).

Author Response

The section "The full-scale vessel" was meant to cover the full-scale propeller behind the hull. But the propeller full scaling was not apropriately explained. This happened because the equations (9) and (10) work well with Kt & Kq curves for full-scale propellers, but this is a claim that I cannot sustanciate within the paper, and ended up leaving the topic unexplained. Thank you for pointing this out.

In the revised paper, formulae for propeller full scaling, following ITTC 7.5-02-03-01.4, has been included. The R2(Ps) for the Series 60 models did not change because the prediction method followed by Todd -from 1933- did not consider full propeller scaling. The R2(Ps) for the KVLCC2 improves and, more significantly, the model is more robust now. A note regarding this has been included right after Table 1.

It is noteworthy to mention that equation (35) does not include the advance ratio, J. This is not a typo. The eq (32) could be expressed as J=f(Rt,deltaCd) with deltaCd=f(...,J), so J can be derived from eq (32) -if Rt and t were known-. It must be done numerically, but in any case, the information captured by J is already contained in the model. 

Also, I prefered expressing the local Reynolds number as shown in eq (34) instead of the Re_07 shown in ITTC 7.5-02-03-02.1 (eq(22) shows that it is the same) because when the model is applied over real operational data, under certain circunstances, the traditional Re_07 makes the model to be numerically unstable. And eq (34) solve these issues.

Thanks again.

Regards,

Javier Zamora

Round 3

Reviewer 2 Report

Dear Author, once again thanks for the applied changes. The paper has been furtherly improved and I don't have major comments.  

Only one minor comment related to reference 54 at Par. 5. This reference (the US patent) seems that is not openly available to the readers. Can the Author quote an open document (or include the relevant points of the procedure directly in this study)? 

Author Response

Please, see attachement.
